# Sustainable Financing and Financial Risk Management of Financial Institutions—Case Study on Chinese Banks

Hao Liu and Weilun Huang *

School of Finance and Trade, Wenzhou Business College, Wenzhou 325035, China
* Correspondence: huangwl@wzbc.edu.cn

**Abstract:** This study examines the relationship between sustainable financing and financial risk management of Chinese financial institutions, using data from Chinese banks. Financial risk management is a comprehensive measure of operating performance, asset quality and capital adequacy ratio. The structural vector auto-regression model determines the relationship between two variables. The positive shock of sustainable financing business negatively impacts the financial risk management of banks. In contrast, positive shock of banks' financial risk management positively affects sustainable financing. Further subdivision of the sample revealed that sustainable financing does not always negatively impact the financial risk management of large state-owned banks. However, the positive shock of financial risk management reduces urban banks' green credit proportions. The results are consistent whenever compared between the empirical outcome of the entire sample and the sample consisting of national joint stock bank accounts. This comparison helps eliminate the possibility of a biased outcome as a major portion of the sample is from a national joint-stock bank account. Apart from data limitations, the results of the sub-sample test are influenced due to the difference in deposit and loan interest rates, as well as different ownership structures of banks.

**Keywords:** sustainable financing; financial institutions; financial risk management

## 1. Introduction

Increased awareness about environmental pollution and natural-resource depletion set the stage for international negotiation about sustainability, nationally determined contribution, and resource-efficient low-carbon circular economy. The outcome of this negotiation can affect an enterprise's long-term economic competitiveness [1–6]. The Paris Agreement, the second international treaty on climate change, established the climate financing mechanism to meet the agreed objective of achieving net-zero within stipulated time through adaptation and mitigation [7]. Accordingly, the Sustainable Development Goal of 2030 highlights the sustainability of water, sanitation, energy, marine resources, ecosystems, and biodiversity [8].

Sustainable financing, or "Environmental Finance" or "Green Financing", becomes critical due to its relationship with enterprise sustainability; it is an investment decision-making process, taking into account both environmental and social aspects [5,6,9,10]. Scholtens [11] believed that environmental finance aims to promote sustainable economic, social, and environmental development through innovations in financial instruments and products. The European Commission [5] suggested that sustainable financing should consider environmental factors, such as climate change, environmental risks, and natural disasters. The financial sector of the EU is following the suggestion, as observed by Eccles and Klimenko [12] and Ahlstrm and Monciardini [10]. In project financing, an assessment of risk based on Equator principles of operational standards plays a crucial role [13]. The United Nations Principles for Responsible Investment provides a direction about environmental and social responsibility in investment decision-making. Ionescu [14] reported significant relationships between green innovation in sustainable finance, corporate environmental

performance, and climate change mitigation. Ionescu [15] opines that green finance should have the potential to promote a low-carbon economic transition, climate change mitigation, and environmental energy sustainability.

Literature reviews and expert surveys reveal that sustainable financing might improve banks' financial risk management, as it could improve their social reputation, social capital, and operating performance [16,17]. Moreover, it can reduce banks' environmental and legal risks [18,19]. Kharlanov et al. [20] showed that during COVID-19, corporate social responsibility reduced an organization's financial risk with commercial profitability. However, it might reduce banks' portfolios because of the crowding-out effect of credit resources [21]. However, sustainable financing might create financial risk for banks due to its fast development and changes in sustainable financing innovations. Morales et al. [22] found that a regulatory framework at national and international levels should be promoted to ensure efficient FinTech (as sustainable financing) governance and adequate industry development. Landi et al. [23] stated that a complete environmental, social, and governance assessment might alter corporate financial risk exposure.

Analysis, identification and control become essential to mitigate potential financial risk before it could threaten their portfolios [24–26]. Popp et al. [24] state that financial risk management is essential for evaluating new agricultural policy instruments. Meyer et al. [26] opine that hydrologic variability poses significant financial risks for hydropower producers, and insurance or risk management instruments (such as sustainable financing in this study) could help manage its financial impacts.

Few studies have discussed the empirical relationship between the sustainable financing of banks and its financial risk management effects. This study explores the area through a case study of 21 banks in China, including the Industrial and Commercial Bank, Agricultural Bank, Bank of China, and Construction Bank (Big Four). The sustainable development strategy to achieve carbon neutrality by 2060 after reaching the peak in 2030 ("30.60") is significant due to China's commitment to the global forum. Banks' sustainable financing is essential to optimize resource allocation, regulate capital flow, and facilitate enterprises to achieve carbon neutrality. The sustainable financing of the Big Four has been raised from Renminbi (RMB) 3.0 trillion to RMB 7.8 trillion from 2016 to 2021, an increase of 39.8% year on year. The proportion of sustainable financing in total loans becomes 11.2% in 2021, investing in clean energy infrastructure, green production system, and conservation of the environment.

By 2021, the Big Four issued 203 "carbon neutral bonds" worth RMB 260.9 billion, equivalent to 42.52% of the green bond issuance in China, to provide direct financing for emission reduction. The China Banking and Insurance Regulatory Commission issued the Green Credit Guidelines in 2012, the first official policy document for sustainable financing in China. However, the absence of appropriate financial instruments for environmental protection, policy paralysis regarding financial incentives, and imperfect legal systems are problems in developing sustainable financing in China [27,28].

This study used the structural vector autoregression (SVAR) model to explore the interaction between banks' sustainable financing and risk management effect. The contributions of this study are as follows. (1) It broadens the concepts of banks' sustainable financing and their financial risk management effect, whereas most literature discusses banks' sustainable financing and their return on equity. This study used comprehensive financial risk management evaluation indices for an in-depth study. (2) It deepens the practical value of banks' sustainable financing and its financial risk management effects through a case study of Chinese banks. Banks with different ownership structures have different product portfolios, management mechanisms, and risk preference policies, which influence the attitude and behavior of banks in developing sustainable financing. (3) The study methodology is unique as SVAR can directly reflect the current influence of the variables in the model and is expected to be better than the VAR method. (4) This study suggests the impacts of banks' sustainable financing and the reasons for different SMEs of banks with different ownership structures.

The remainder of this paper is organized as follows. Section 2 describes the measurements of banks' sustainable financing and its financial risk management effects; Section 3 constructs a theoretical hypothesis to explain the relationship between banks' sustainable financing and financial risk management effects; Section 4 shows the empirical analysis; Section 5 presents the discussion; and Section 6 concludes the paper and provides policy suggestions.

## 2. Development and Measurements of Chinese Banks' Sustainable Financing and Financial Risk Management Effects

Sustainable financing is a way to respond to the impact of COVID-19 policies in some countries, focusing on the support of green investment [29]. Nandiwardhana and Dan [30], Huang et al. [31], and Yuan and Gallagher [32] stated that banks' primary functions in sustainable financing are financial intermediation and distribution that would be beneficial to the sustainable development of a country in terms of development innovations, building social strength, and defenses against disasters. Vagin et al. [33] find that financial risk management with corporate social responsibility might be helpful for sustainable development. Mejia-Escobar et al. [34] found that sustainable financial products in the Latin American banking industry could be classified as financing or refinancing destinations, total or partial subjects, and environmental, social, or mixed benefits. Social-oriented sustainable financial products are infrastructure, housing, employment, microfinance, food security, socioeconomic advancement, and empowerment. The categories for green-oriented sustainable financial products are energy, climate, pollution, natural resources, and biodiversity.

Sustainable financing for the economic and social development of China involves financial tools (loans, bonds, and insurance), credit rating standards, performance evaluation of financial institutions, a listing of qualified enterprises, and the involvement of the international sustainable financing market [35]. Interestingly, most studies have examined the credit effect of sustainable financing on the economy and market participants, covering fields such as industrial structure upgrading [36], sustainable corporate financing behavior [37], and financial risk management of banks [38].

The study surveyed the sustainable financing data of 21 Chinese banks from 2014–2020, as given in Table 1, to discuss the development and measurement of Chinese banks' sustainable financing. From Table 1, the total loan amount of major Chinese banks' sustainable financing increased significantly from 2014 to 2020, from RMB 6.01 trillion to RMB 11.5 trillion, with an average annual growth rate of 11.42%. Sustainable financing activities in China include loans for energy conservation, environmental protection and manufacturing activities of three strategic industries: energy conservation, environmental protection, and new energy vehicles. Though a survey of experts between 2018–2020 does not give any empirical evidence, the loans for energy conservation and environmental protection are the maximum among all the projects.

**Table 1.** Sustainable Financing of 21 Chinese Banks in 2014–2020. Unit: 1 trillion RMB.

| Year | Strategic Emerging Industries Loans | Energy Conservation and Environmental Protection Projects Loans | Total Loans |
|---|---|---|---|
| 2014 | 1.58 | 4.44 | 6.01 |
| 2015 | 1.69 | 5.32 | 7.01 |
| 2016 | 1.70 | 5.81 | 7.50 |
| 2017 | 1.76 | 6.53 | 8.30 |
| 2018 | - | - | 9.66 |
| 2019 | - | - | >10.00 |
| 2020 | - | - | 11.50 |

The variable to understand the sustainable financing performance of Chinese banks is Green Loan Proportion (GLP). A survey revealed that loan proportions of listed Chinese banks favoring sustainable financing are less than 10% of total loan disbursement, the

only exception being China Industrial Bank, with more than 10% implemented as equator principles. The proportion of sustainable financing in China Industrial Bank also reached 30.22% in 2019. The influencing factors of GLP might be related to regulations and policies of the government. China has incorporated environmental protection into credit risk assessment regulations since 2007. Since 2013, a statistical system for sustainable financing has been formally established to understand banks' sustainable financing status. Since 2018, banks' sustainable financing performance has been incorporated into their macro-prudential evaluation system. However, most Chinese banks are not members of the equator standard banking system. Their business strategies may not focus on sustainable financing, as it may affect their profits.

The government, along with the Central Bank, is responsible for providing a policy framework for developing sustainable financing. As a regulatory authority, the government is also responsible for corrective punitive action in case nonperforming assets of Chinese banks increase, thus, affecting the objective of sustainable financing [30,39,40].

This study used banks' capital adequacy ratio, asset quality, earnings, and liquidity from the CAMEL rating system to evaluate Chinese banks' financial risk management performance. These four indicators indicate the comprehensive financial risk management of the Chinese banks, including their operational performance and risk management ability. Financial risk management indicators, formulae, and regulatory requirements used in this study are given in Table 2. The CAMEL rating system provides the banks' comprehensive risk management abilities, used for internal control, operation, and risk management [41–43]. Sahyouni et al. [44] considered the CAMEL rating system as an indicator of bank strength. Nguyen and Dang [45] used the CAMELS rating system to verify the bank-specific determinants of Vietnam's loan growth. Kliestik et al. [46] found that the ratios of current assets, total liabilities-to-total assets, and total sales-to-total assets are frequently used variables to predict a country's financial distress.

**Table 2.** Chinese Banks' Financial Risk Management indicators, calculation formula, and regulatory requirements.

| First-Level Index | Second-Level Index | Calculation Formula | Regulatory Requirements (%) |
|---|---|---|---|
| Profitability | Return on total assets (ROA) | Net profit/average total assets | - |
| | The weighted average return on equity (ROE) | Net profit/weighted average net asset | - |
| Capital adequacy | Capital adequacy ratio (CAR) | Capital/risk-weighted assets | 10.5 |
| | Tier 1 capital adequacy ratio (TCAR) | Tier 1 capital/risk-weighted assets | 8.5 |
| Asset quality | Nonperforming loan ratio (NPLR) | Nonperforming loans /total loans | 5 |
| | Performing Loan Ratio (PLR) | Loan impairment provision/nonperforming loans | 150 |
| Liquidity | Liquidity ratio (LR) | Current assets/current liabilities | 25 |
| | Loan-to-deposit ratio (LDR) | loans/deposits | 75 |

**Note:** Regulatory requirements may vary depending on the bank's structure or from time to time and are listed in this table as reference standards at the time of analysis (2019).

The indicators of the CAMEL rating system include capital adequacy, asset quality, management, earnings, liquidity, and sensitivity to market risk. The following are reasons

for using the CAMEL rating system in this study: (1) The evaluation of a bank's management is a qualitative analysis that requires the regulatory authorities to assess the bank's daily situation comprehensively. No such data is available for Chinese banks. (2) The evaluation of a bank's market risk sensitivity is relatively complex, and there are still no public data available on Chinese banks. (3) Neither the bank's management nor market risk sensitivity is related to the financial indicators. After careful evaluation of contemporary literature, the author uses return on equity (ROE) to evaluate banks' profitability, capital adequacy ratio (CAR) for banks' capital adequacy, nonperforming loan ratio (NPLR) and performing loan ratio (PLR) for banks' asset quality, and liquidity ratio (LR) and loan-to-deposit ratio (LDR) for banks' liquidity as banks' return on assets (ROA) is highly correlated with its ROE and its CAR is with its Tier 1 capital adequacy ratio (TCAR).

Data from 15 listed banks between 2008 and 2019 were used in this study to compute descriptive statistics and trend analysis based on the framework of evaluation indicators of sustainable financing and financial risk management (Table 3). The research objects in this study are large Chinese state-owned banks, national joint-stock banks, and urban banks. Large state-owned banks are Industrial and Commercial Bank of China, Agricultural Bank of China, China Construction Bank, Bank of China, and Bank of Communications. National joint-stock banks include China Merchants Bank, China Industrial Bank, China CITIC Bank, China Everbright Bank, Shanghai Pudong Development Bank, Ping An Bank, and Huaxia Bank. Urban banks are Bank of Beijing, Nanjing Bank, Bank of Ningbo, and Jiangsu Bank, listed companies. There are 145 sample observations from 2008 to 2019. As some banks' sustainable financing data in some years have not been published clearly, the sample is unbalanced panel data. The symbols of banks used in Table 3 are as follows: The Industrial and Commercial Bank of China is 111, Agricultural Bank of China is 112, China Construction Bank is 113, Bank of China is 114, Bank of Communications is 115, China Merchants Bank is 116, China Industrial Bank is 117, China CITIC Bank is 118, China Everbright Bank is 119, Shanghai Pudong Development Bank is 120, Ping An Bank is 121, Huaxia Bank is 122, Bank of Beijing is 123, Nanjing Bank is 124, Bank of Ningbo is 125, and Jiangsu Bank is 126.

**Table 3.** Descriptive Statistics of Chinese Banks' Sustainable Financing and Financial Risk Management Indicators.

|  | **GLP** | **ROE** | **CAR** | **NPLR** | **PLR** | **LR** | **LDR** |
|---|---|---|---|---|---|---|---|
| Mean | 5.33 | 17.49 | 12.73 | 1.22 | 251.67 | 46.88 | 72.28 |
| Median | 4.18 | 17.27 | 12.41 | 1.18 | 228.20 | 46.75 | 71.62 |
| Maximum | 30.22 | 27.41 | 17.52 | 2.65 | 524.08 | 75.07 | 109.45 |
| Minimum | 0.37 | 10.61 | 9.88 | 0.38 | 121.72 | 27.60 | 45.98 |
| Standard deviation | 5.10 | 3.66 | 1.58 | 0.46 | 94.33 | 9.55 | 11.26 |

Based on literature review and expert interviews, this study develops a comprehensive evaluation index on the indicators mentioned above and their weights, which helps to evaluate the order of each bank scientifically. The weights of the evaluation indicators in this study were 30% for ROE, 20% for CAR, 15% for NPLR, 15% for PLR, 10% for LR, and 10% for LDR.

The value of each index (Figure 1a–h) varies significantly among banks. The ROEs of Chinese banks between 2008 and 2019 showed downward trends; their CARs fluctuated considerably, decreasing from 2012, and capital was supplemented from 2017. Their NPLRs began to increase since 2012 and remained high. Conversely, their PLRs began to decrease in 2012, followed by an increase in 2017. Their LR changes irregularly but generally in accordance with China's regulatory requirements. LR started improving since 2017 due to their strong liquidity management. With the Chinese government's relaxation of LDR supervision in 2015, banks' LDRs have rapidly diverged. However, comprehensive financial

risk management of Chinese banks shows regularity. There was an overall upward trend from 2009 to 2012, a downward trend from 2013 to 2017, and a subsequent rise after 2017.

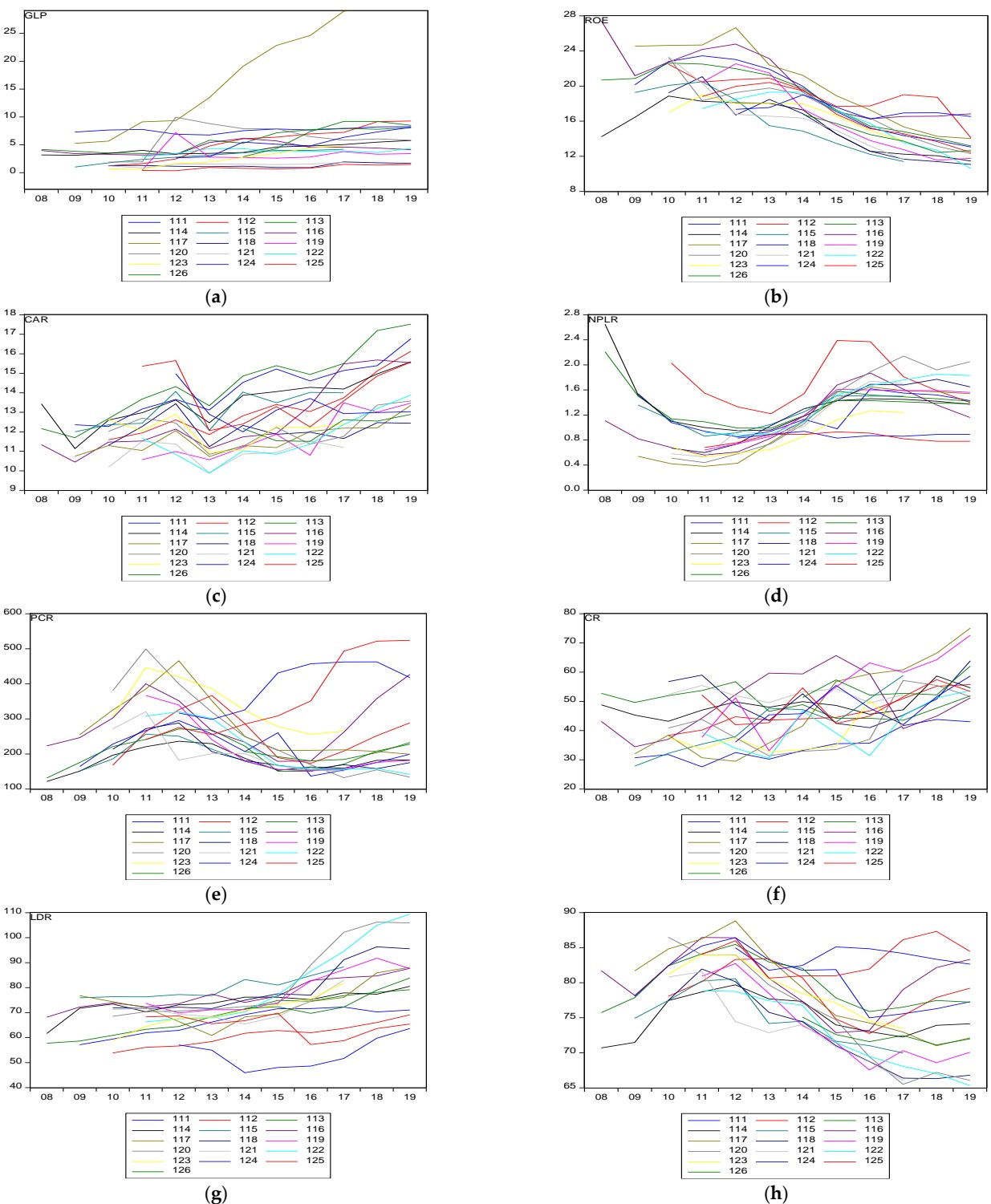

**Figure 1.** Trend Graphs of Chinese Banks' Sustainable Financing and Financial Risk Management Indicators. (**a**) GLP of Chinese banks in 2008–2019. (**b**) ROE of Chinese banks in 2008–2019. (**c**) CAR of Chinese banks in 2008–2019. (**d**) NPLR of Chinese banks in 2008–2019. (**e**) PLR of Chinese banks in 2008–2019. (**f**) LR of Chinese banks in 2008–2019. (**g**) LDR of Chinese banks in 2008–2019. (**h**) The Comprehensive Financial Risk Management of Chinese banks in 2008–2019.

### 3. The Theoretical Reciprocal Causation Relationships of Banks' Sustainable Financing and Financial Risk Management

A theoretical study of the impacts of banks' sustainable financing on financial risk management (Figure 2) indicates an ROA reduction in the short run and an improvement of banks' asset quality and profit in the long run. Thus, it enhances banks' financial risk management performance [18,19].

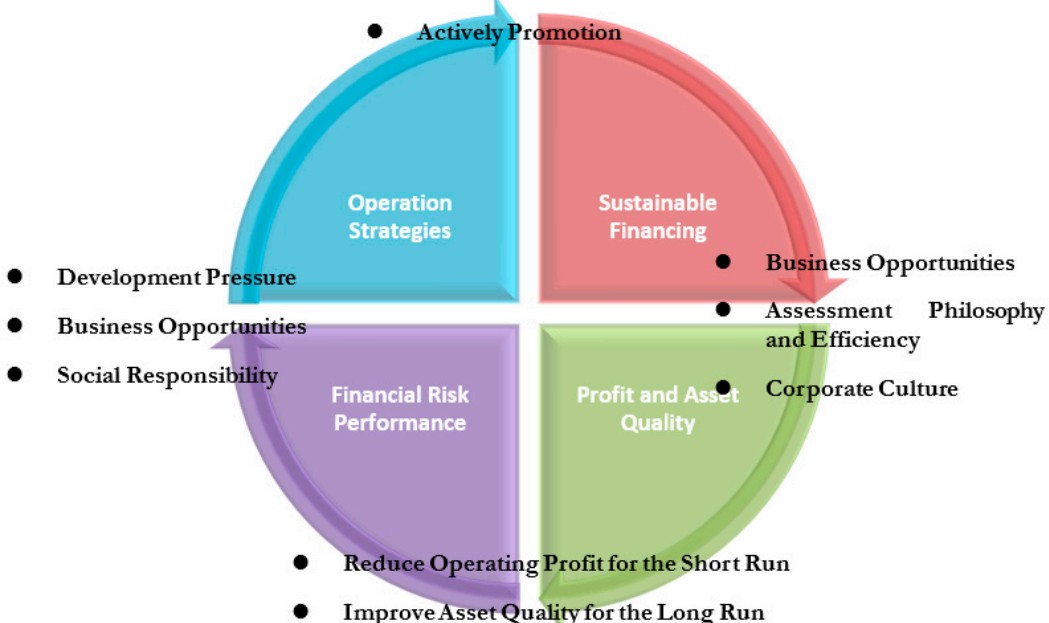

**Figure 2.** The Theoretical Reciprocal Causation Mechanisms of Banks' Sustainable Financing and Financial Risk Management.

The mechanism by which banks' sustainable financing affects profit and asset quality are as follows:

(1) Business opportunities and risky project prevention: Banks can increase market share and profitability by expanding sustainable financing activities for governments and investors. Investments in energy conservation, environmental protection, and strategic industries have rapidly developed. However, banks with more resource-oriented sustainable financing businesses can better resist the economic cycle, as most governments would promote sustainable financing policies to realize economic growth. Due to regulatory restrictions and budget constraints, banks could stop providing loans to industries with high pollution, higher energy consumption, and with an excess capacity; for example, steel, paper, aluminum, and flat glass industries.

(2) New credit evaluation policies and evaluation efficiency: Banks' sustainable financing activities may increase operational costs in the short run due to the implementation of new evaluation policies and time-gap for learning the new process and compliance with environmental policies and laws.

(3) Corporate culture with risk management: This is an important part of internal control mechanism due to its behavioral standards that could help a bank to control risks and ensure steady development [47,48]. As projects under sustainable financing policies focus on government, banks with sustainable financing projects might acquire subsidies from the government; thus, justifying the need to improve risk management capacities while complying with sustainable financing requirements.

Theoretical understanding of the impact of efficient financial risk management indicates (Figure 2) more sustainable financing activities, confidence about a portfolio of

high-risk and high-yield assets, better profitability, and capacity to sustain sustainable financing activities in the long run.

Accordingly, we propose the following hypotheses:

**Hypothesis 1 (H1).** *Banks' sustainable financing has a significantly positive effect on their financial risk management.*

**Hypothesis 2 (H2).** *Banks' financial risk management has significantly positive effects on sustainable financing.*

## 4. The Empirical Reciprocal Causation Relationships of Banks' Sustainable Financing and Financial Risk Management

### 4.1. The Model

This study used the panel SVAR model to examine the relationship between banks' sustainable financing and comprehensive financial risk management [49]. SVAR can address the deficiency of the VAR model by indicating the current influence of the variables. It considers the impulse-response function, an average reflection of endogenous variables on exogenous shocks, and immediate effects among the variables. Esmaeili and Rafei [50] applied the SVAR model to analyze the factors affecting fluctuations in electricity consumption based on economic conditions. Kse and Nal [51] used the SVAR model to study the effects of oil prices and their volatility on inflation in Turkey. Sarwar et al. [52] used SVAR and the time-varying parameter vector autoregressive model to analyze the relationships among electricity consumption, oil revenues, GDP, and inflation shocks.

The assumptions of the SVAR model are as follows:

(1) The difference between the proportions of banks' sustainable financing in periods t and $t - 1$ is set as $DGLP_t$, and the difference between banks' comprehensive financial risk management in periods t and $t - 1$ is set as $DFS_t$. Thus, $y_t = (DGLP_t, DFS_t)'$.

(2) The vector of the exogenous control variable is set $x_t$, where $x_t = (Dlnsize_t, DCIR_t, DlnGDP_t)'$. $Dlnsize_t$ is the difference in the banks' asset size logarithms in periods t and $t - 1$, and its control reason is the differences in its financial impact based on the bank's asset scale. $DCIR_t$ is the difference in banks' cost–income ratios in periods t and $t - 1$, where the cost–income ratio represents its operational capability. $DlnGDP_t$ is the difference in GDP logarithms in periods t and $t - 1$, the impact of which differs depending on the economic cycle.

(3) $\Gamma$ is the lag operator and p is the lag order. Matrix A reflects the current interaction between $DGLP_t$ and $DFS_t$; $u_t$ is the disturbance term of the reduced form; $E_t$ is the disturbance term of structural form; and the covariance matrix of $\varepsilon_t$ is normalized as a unit matrix, assuming it follows a multi-dimensional normal distribution. B is a $2 \times 2$ matrix.

(4) Under the condition of differences in endogenous variables, the fixed effect and invariant variables over time for each sample are effectively controlled. Thus, the AB-type SVAR model was constructed as follows:

$$Ay_t = \Gamma_1 y_{t-1} + \Gamma_2 y_{t-2} + \cdots + \Gamma_p y_{t-p} + Cx_t + u_t \tag{1}$$

$$A\{I - \Gamma_1 L - \cdots - \Gamma_p L^p y_t - Cx_t\} = Au_t = B\varepsilon_t \tag{2}$$

This model analyses the dynamic interaction between sustainable financing and banks' comprehensive financial risk management by imposing short-term constraints on matrices A and B. The response of the variables to shock is presented mainly through the structural impulse response function.

The empirical study shows that banks' sustainable financing impacts their financial risk management (Figure 3). There is a weak negative correlation between the average annual growth rate of banks' sustainable financing ($DGLP_t$) and their comprehensive financial risk management ($DFS_t$) in period t (Figure 3). The average annual growth rate

of Chinese banks' sustainable financing proportion was 0.50%, with a standard deviation of 1.40%.

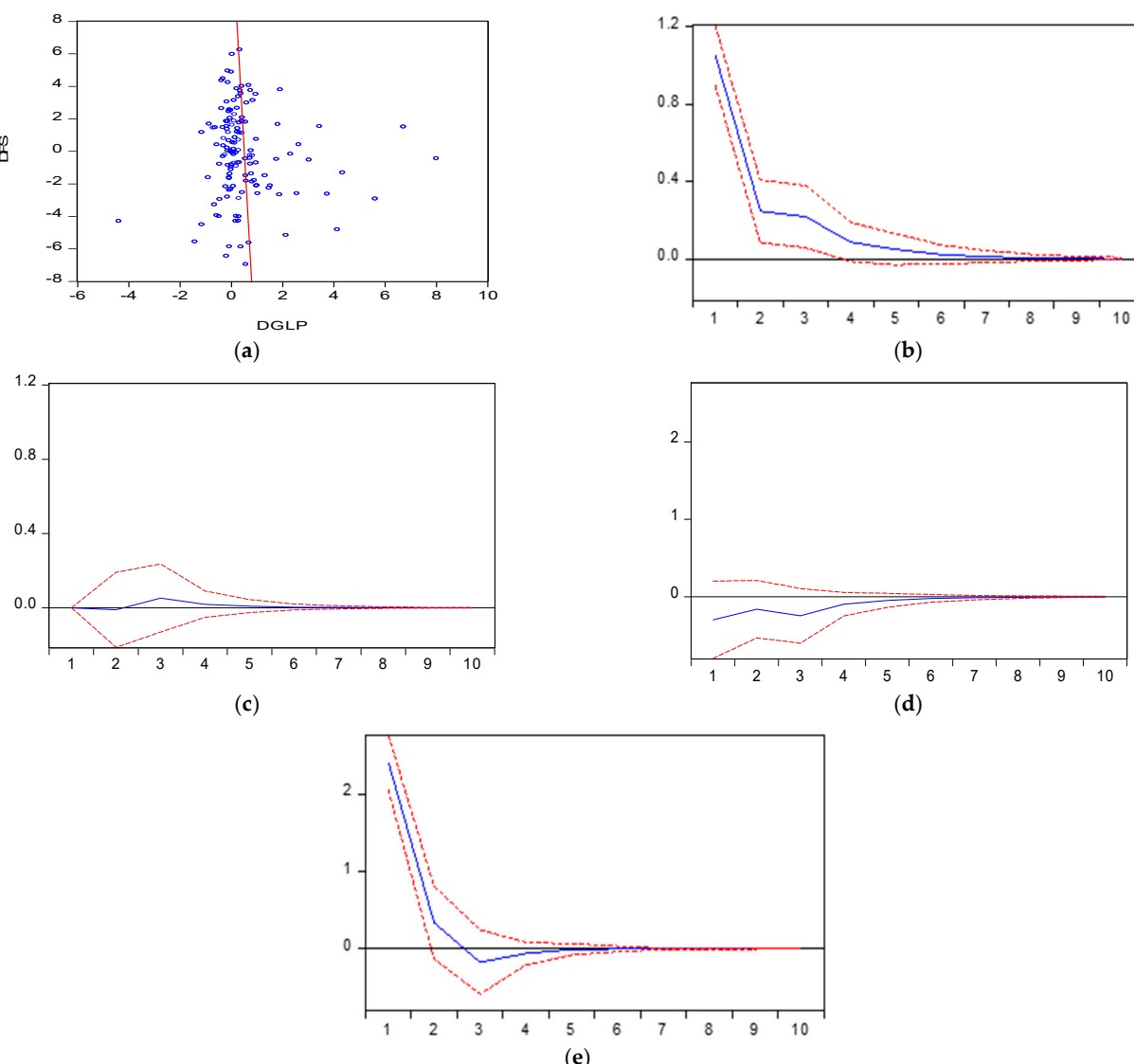

**Figure 3.** The Correlation and Impulse Response Function of Chinese Banks' Sustainable Financing and Financial Risk Management Indicators. (**a**) The Correlation between DGLP and DFS of Chinese banks in 2008–2019. Its blue dots are sample points, its red line is trend line. (**b**) Response of DGLP to DGLP. (**c**) Response of DGLP to DFS. (**d**) Response of DFS to DGLP. (**e**) Response of DFS to DFS. Their blue lines in (**b**–**e**) are the lines of average response, its red lines in (**b**–**e**) are the lines of average response ± two standard deviations.

From the results of the pulse reflection functions in Figure 3, there might be a significant negative time-delay effect of banks' sustainable financing on their financial risk management. However, banks' financial risk management may have significant positive effects on their sustainable financing. As seen in the above figure, by the Cholesky decomposition innovation method, if there is a one-unit exogenous increment of banks' financial risk management in the first year, their sustainable financing ratio will reach a peak of 0.05 in the third year. However, if there is a one-unit exogenous increment of banks' sustainable financing ratio in the first year, their financial risk management would decrease by 0.3 in the first year. Banks' sustainable financing activities would affect their profitability in the short run, but efficient financial risk management performance could encourage further engagement in sustainable financing activities while paying attention to social

responsibility. Therefore, the impact of sustainable financing on financial risk management might emerge in the current period, while the impact of financial risk management on sustainable financing might have a time-delayed effect.

From the above discussion, H1 and H2 are supported in the short run but might not be supported in the long run.

### 4.2. Empirical Results

The empirical results show that the panel SVAR model is stable, and the impact-effect converges to zero in the long run. It indicates that the shock of banks' sustainable financing ratio on financial risk management is temporary, and endogenous variables behave according to the established trend in the long run. To avoid possible pseudo-regression problems, caused by time-series regression, the ADF test was conducted for a unit root of variable, and the results showed that all variables show stationarity at a significance level of 10%.

If the scale of banks' sustainable financing is too low compared to overall business development, it indicates apathy of bank management for sustainable financing to minimize the financial risk and maximize profit. As financial innovations would focus on profit alone, it suppresses banks' willingness to finance a sustainable financing project, thus affecting initiative for sustainable development. Compared to sustainable financing, the financial restriction on industries with high pollution and energy consumption might influence banks' financial risk management more significantly.

For the accuracy of interpretation, the result is presented according to the types of banks, namely, state-owned banks, joint-stock banks, and urban banks (Figures 4–6), to comply with the homogeneity assumption of SVAR estimation, ownership relations, management mechanisms, asset scales, sustainable financing activities, and financial risk management performances [19].

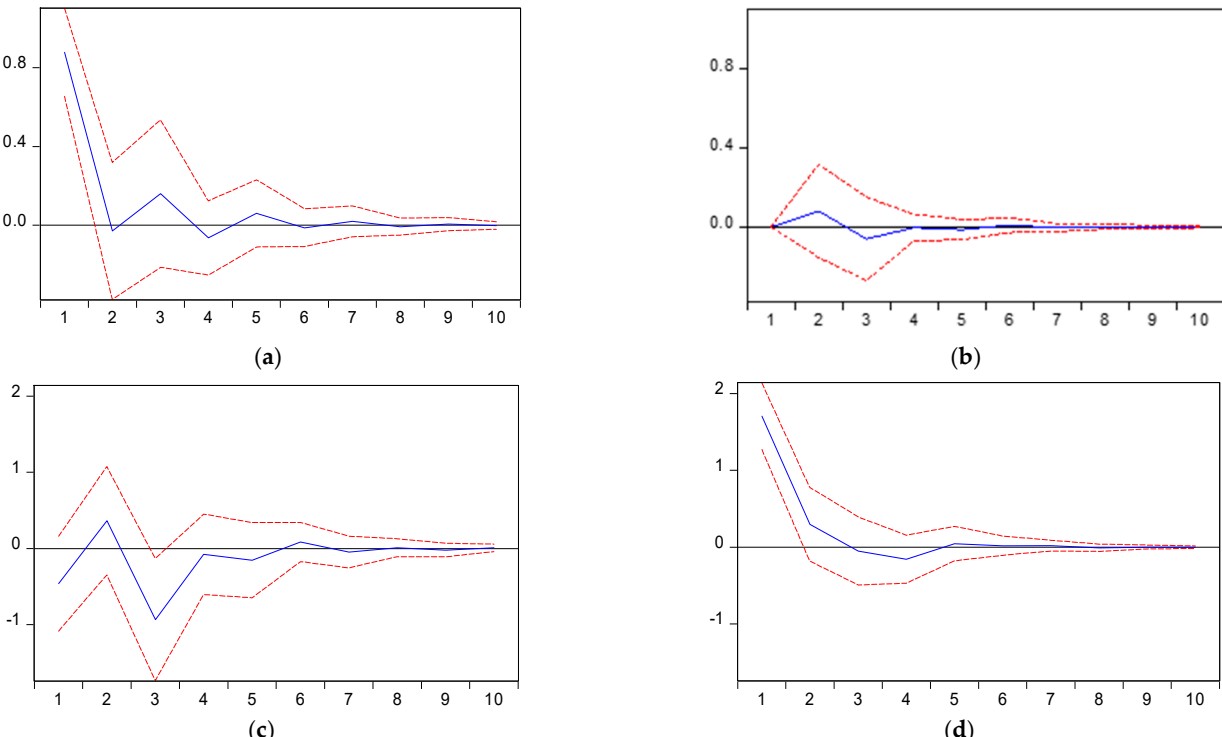

**Figure 4.** The Impulse Response Function of Chinese State-owned Banks' Sustainable Financing and Financial Risk Management Indicators. (**a**) Response of DGLP to DGLP in State-owned Banks. (**b**) Response of DGLP to DFS in State-owned Banks. (**c**) Response of DFS to DGLP in State-owned Banks. (**d**) Response of DFS to DFS in State-owned Banks. Their blue lines in (**a**–**d**) are the lines of average response, its red lines in (**a**–**d**) are the lines of average response ± two standard deviations.

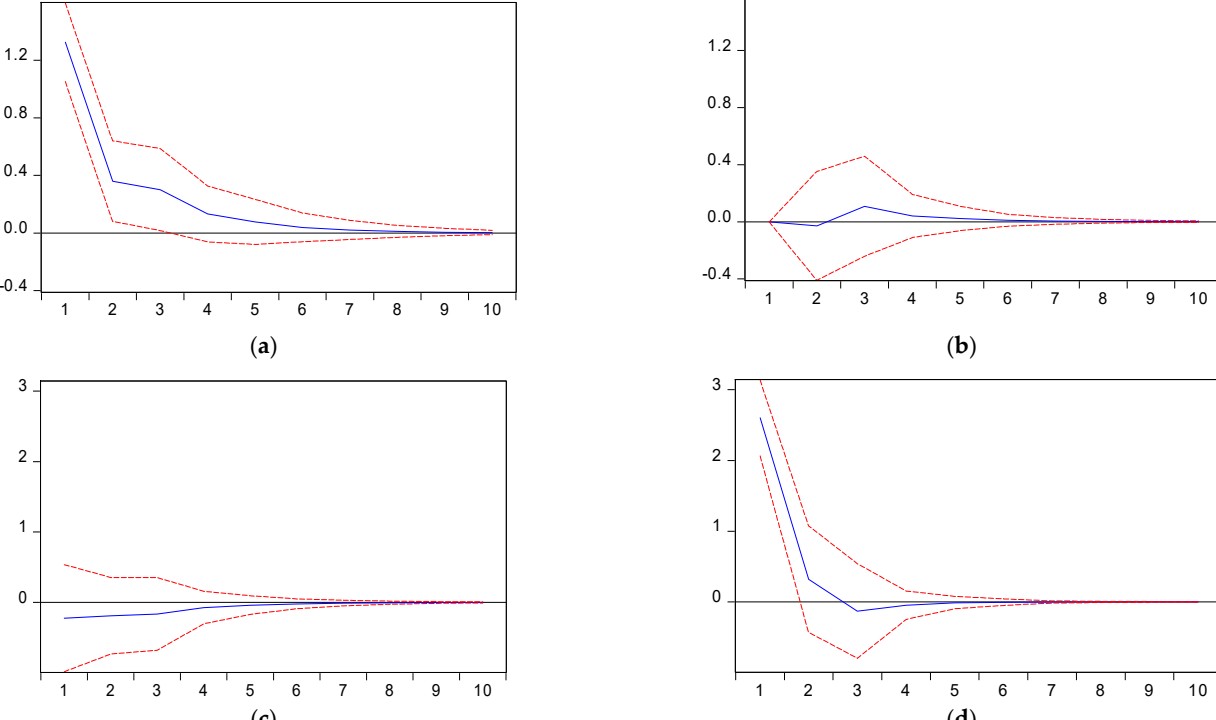

**Figure 5.** The Impulse Response Function of Chinese Joint-Stock Banks' Sustainable Financing and Financial Risk Management Indicators. (**a**) Response of DGLP to DGLP in Joint-stock Banks. (**b**) Response of DGLP to DFS in Joint-stock Banks. (**c**) Response of DFS to DGLP in Joint-stock Banks. (**d**) Response of DFS to DFS in Joint-stock Banks. Their blue lines in (**a**–**d**) are the lines of average response, its red lines in (**a**–**d**) are the lines of average response ± two standard deviations.

The results indicate (Figures 3 and 4) that the effects of state-owned banks, in the context of sustainable financing and risk management, are different from all other banks. There might have been alternatingly positive before the adverse effects of state-owned banks' sustainable financing on their financial risk management (Figure 4). The positive shock on the proportion of sustainable financing business in state-owned banks negatively impacts financial risk management in the current year and positively impacts the following year.

The impulse response function of the sample of national joint-stock banks in Figure 5 shows almost the same characteristics as those of the full sample. The positive shock to the banks' financial risk management will increase the proportion of sustainable financing from the third year. The better the financial risk management of joint-stock banks, the more motivated it will be to engage in sustainable financing in the future. Consistent with the entire sample, sustainable financing has a negative impact on banks' financial risk management.

The impulse response function of the sample of urban banks (Figure 6) shows a significantly different phenomenon from that of the national joint-stock banks. The financial risk management of urban banks reduces the proportion of sustainable financing. In effect, the better the financial risk management of urban banks, the less incentive they have to engage in sustainable financing. This phenomenon is not surprising, as urban banks have a strong incentive to make more profit and occupy higher ranks; management usually pays little attention to sustainable financing business.

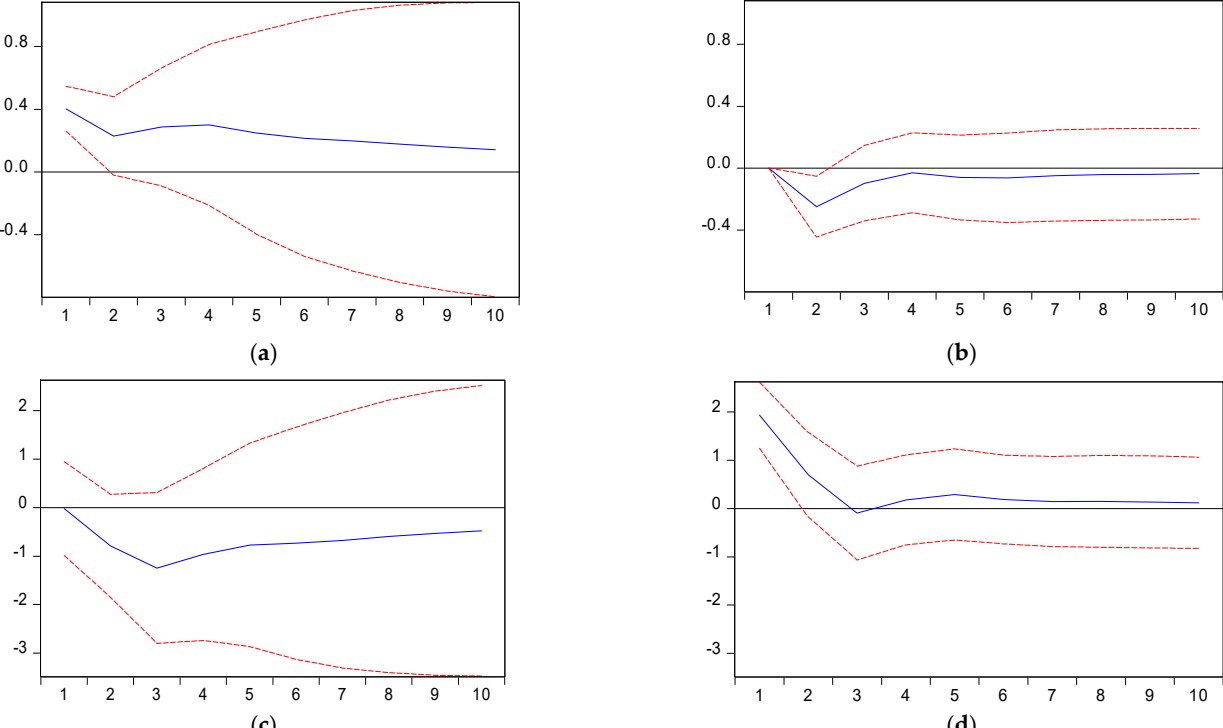

**Figure 6.** The Impulse Response Function of Chinese Urban Banks' Sustainable Financing and Financial Risk Management Indicators. (**a**) Response of DGLP to DGLP in Urban Banks. (**b**) Response of DGLP to DFS in Urban Banks. (**c**) Response of DFS to DGLP in Urban Banks. (**d**) Response of DFS to DFS in Urban Banks. Their blue lines in (**a**–**d**) are the lines of average response, its red lines in (**a**–**d**) are the lines of average response ± two standard deviations.

The reason for the results mentioned above could be the different ownership of state-owned, joint-stock, and urban banks. The least costly state-owned banks that accept deposits from their customers used to provide loans with a low-interest rate. Interestingly, financing a sustainable business has little impact on its profits; however, increasing the quality of its assets is always beneficial. Therefore, developing a sustainable financing business is likely to improve financial risk management. Joint stock banks and city banks have a high cost of absorbing deposits. Their customers are often small- and medium-sized enterprises or restricted high-risk industry customers. Loan interest rates are generally high in such cases. The development of low-interest-rate sustainable financing businesses might reduce their operating profits and have a negative impact on financial risk management. However, in recent years, China's joint-stock banks have developed rapidly, the asset scale has gradually approached large-scale state-owned banks, and their sense of social responsibility has gradually strengthened.

*4.3. Robustness Test*

According to our theoretical analysis, the development of a bank's sustainable financing business is an important external performance of the conservative risk-management culture of banks. Although a sustainable financing business has low-risk characteristics, it is also a low-income business. Therefore, the relationship between the comprehensive index of banks' financial risk management and the proportion of sustainable financing may be uncertain and may depend on the weight of profitability and asset quality. Therefore, we chose to change the measurement of the financial risk management evaluation index to analyze the robustness of the empirical model. Specifically, we eliminate the profitability index, distribute its weight evenly with other indicators, and reconstruct a comprehensive commercial bank financial risk management evaluation index. The new estimation results also show that the positive shock on banks' financial risk management will promote the

development of banks sustainable financing business. The positive impact of a sustainable financing business comes with a negative effect on banks' financial risk management.

After replacing the variables, the exogenous shock on one unit of banks' financial risk management will promote the proportion of sustainable financing business to increase by 0.06 in the third year, which is slightly higher than the original SVAR model result of 0.05. The exogenous impact of sustainable financing accounting for one unit affects the development quality of the current banking industry by 0.3, which is the same as the original SVAR model. However, variable replacement increased the standard deviation (confidence interval) of the peak response.

## 5. Discussion

A comparison with other literature reveals that most studies found a positive relationship between sustainable financing and financial institution performance. According to the current author, this might be true in the short run but not in the long run, especially for financial risk management. Moreover, the relationships between sustainable financing and risk management differ according to the shareholding structures of the organization.

Due to article length limitations, this study did not incorporate assumptions regarding some regulations and supervisions. Future studies should focus on addressing the following limitations faced in this study:

(1) There are no discussions on the importance of government regulations. Banking regulations and the government's supervision of sustainable financing and financial risk management should play critical roles in realizing the objectives of sustainable financing and financial risk management. Smoleńska and van't Klooster [53] proposed that regulators would face political choices of climate policies and micro-prudential framework for banks, and lawmakers would consider their issues of legality, legitimacy, and accountability.

(2) The study does not discuss the target industry or enterprises for sustainable financing. Financial institutions can promote their financial risk management through their lending decisions and policies. Cheung et al. [54] argued that the barriers to sustainable financing are economic markets, political institutions, and socio-cultural behavior. Ascui et al. [55] suggested that financial institutions are indirectly exposed to risks due to their dependence on natural capital and ecosystem services.

(3) The study does not discuss sustainable financing and financial risk management disclosure. The continuous and full disclosure of sustainable financing and financial risk management could improve banks' sustainable development, as banks' economic and financial disclosures are widely disclosed information, while environmental and risk management disclosures are relatively less disclosed Gunawan et al. [56] find that sustainability and green banking disclosures are important to the banking sector.

## 6. Conclusions

This study discussed the correlation mechanism between banks' sustainable financing and financial risk management. By constructing the SVAR model, it was found that, overall, the positive shock on sustainable financing business has a negative impact on banks' financial risk management. In contrast, the positive shock on banks' financial risk management positively affects sustainable financing. After further subdivision of the samples, we find that sustainable financing does not always hurt banks' financial risk management for large state-owned banks. The shock to sustainable financing promotes the financial risk management of state-owned banks in the second phase. Urban banks' positive shock to their financial risk management reduces the proportion of sustainable financing. As national joint-stock banks account for most of the samples, the research conclusions of national joint-stock banks are consistent with the empirical results of the whole sample.

It can be predicted that the sustainable financing business will also receive increasing support from banks with the transformation and upgrade of China's economic structure, the

rapid development of the new energy automobile industry, and other environment-friendly manufacturing industries. Though the proportion of sustainable financing may remain low in Chinese banks, the significance of sustainable financing cannot be ruled out due to the paucity of data, and credit approval for green development is crucial. The enhancement of environmental awareness, risk avoidance of environmental laws and regulations, and the remolding of prudent management culture should become the essential basis and content for the high-quality development of banks. Therefore, the relationship between financial institutions' social responsibility and sustainable financing is an important future research direction.

As engaging in sustainable financing has a negative impact on banks' financial risk management, they lack the motivation to engage in sustainable financing. It is necessary to improve and enhance banks' motivation and business capability for sustainable financing from the aspects of the legal system, Government and public restraint, supervision guidance, and talent cultivation.

The first policy recommendation is to improve the legal system for green development by providing a sound legal framework for the development of sustainable financing. The experience of sustainable financing in a developed country indicates that the development of sustainable financing is inseparable from the legal systems. The first level is environmental protection laws and regulations, which are the basic premises and driving forces for the development of sustainable financing. The legal system for environmental protection has already been framed in developed countries, and the economic responsibilities are clearly stipulated in the environmental protection laws. Environmental laws focus on the protection of natural resources, but not on preventing environmental pollution. Moreover, environmental laws, with poor operational feasibility are difficult to enforce. Moreover, Chinese law never mentions the responsibilities of financial institutions, and thus, researchers are advocating sound financial law and environmental legislation for China.

The second policy recommendation is to build a diversified environmental governance system for China's sustainable development, with the involvement of other stakeholders, such as social organizations. Co-construction, sharing, and co-governance could facilitate policy implementation and resource allocation through a market mechanism. Under the multidimensional green governance system, financial institutions could consciously incorporate environmental factors into the business evaluation process, decide about risk and return before extending loans, and thus, play a crucial role in fulfilling the nation's sustainable development objective.

The third policy recommendation is to establish a green financial market system to speed up the green development initiative of financial institutions. The "Equator Principles" can act as a guideline for the banking industry to develop sustainable financing. State-owned banks and financial regulatory authorities may come forward with regulatory guidance to fulfil green development initiatives.

The fourth policy recommendation is to enhance support services and the professional ability of financial institutions to carry out sustainable financing. Strengthening green financial service support can effectively improve the efficiency of financial institutions in carrying out green financial services, improve the ability of green financial services for green development, and effectively prevent financial risks. Supportive services include green financial intermediaries and green financial talent.

The fifth policy recommendation is to strengthen risk prevention and control of sustainable development. Combining traditional risk control and intelligent risk control forms a closed-loop green financial risk prevention and control system from source to end. The first step is to prevent the credit risk of green projects from the source, implement environmental protection standards and credit risk management requirements for customer investigation, credit review, post-loan management, and other links, and establish the entire process of the green financial risk control system. The second is to introduce professional third-party green assessment institutions, improve the green financing review mechanism, conduct a strict risk assessment of green projects, and track the progress of green projects

in real-time. Third, improving the intelligent risk prevention and control system, making full use of big data, cloud computing, artificial intelligence, and other financial technology means establishing a more accurate intelligent risk control management system, improving the level of risk pricing, tracking user credit data, and monitoring user post-loan behavior. Fourth, environmental risk stress tests should be carried out as soon as possible, along with studying the methods and experience of banks, establishing a stress test process with carbon price, emission reduction technology, and other vital factors as stress factors, predicting the impact of industrial transformation on bank credit risk in advance, and developing an early warning system.

The sixth policy recommendation is that the issues of sustainable financing in China should expand to include social considerations, such as inequality, inclusiveness, labor relations, investment in human capital, and communities [5]. Banks should deeply integrate sustainable financing with technological, inclusive, and consumer financing to complement each other's advantages and develop in a staggered way.

The seventh policy recommendation is that the policy applicability of a new Chinese sustainable financing tool (crowdfunding) should be studied. Ari and Koc [57] discussed the advantages of the philanthropic-crowdfunding-partnership model renewables which is a solution for financing regional sustainable development considering society and the environment.

To develop sustainable financial industry, we must strengthen the service capacity of green financial intermediary service institutions. At present, China's financial institutions are faced with the constraints of their ability and professional levels to develop sustainable financing businesses. As environmental protection involves a wide range of fields, including green production technology, environmental prevention, and control technology, current law and accounting firms in the financial market do not have the excellent capability of green financial intermediary services. Therefore, we should establish an extraordinary evaluation institution for green financial products and services to facilitate the development of sustainable financing in China.

It is also necessary to cultivate several green financial talents as support. With the deepening of the green development concept and green financial business, the number of Chinese banks joining "Equator Principles" will gradually increase. Sustainable financing, green bonds, green insurance, green funds, and other green financial instruments are constantly growing. The development of sustainable financing is bound to face a shortage of talent familiar with environmental laws, policies, and environmental risk assessment. Therefore, financial and social education institutions need to make joint efforts to strengthen the cultivation of green financial talent to provide strong human capital support for the green development of China's economy and financial institutions.

The data limitations of this study are as follows: (1) The period limitation of the observable bank data is mostly from 2011–2019. The impacts of the average annual growth rates of banks' sustainable financing and their comprehensive financial risk management might be underestimated because the optimal lag order is limited. (2) Most banks' sustainable financing proportions are too low to efficiently estimate their impacts on their comprehensive financial risk management.

Future studies should focus on the reverse impact of a bank's financial risk-management performance on its sustainable financing performance. As Nandiwardhana and Dan [30] suggested, banks might have a negative impact on sustainable development during their business strategies without financial risk management. Future studies could improve the bank ranking process using the fuzzy analytic hierarchy process, and fuzzy technique for order preference by similarity to an ideal solution [58], as Varga et al. [59] used CAMEL and similarity analysis methods to analyze the performance of the Turkish Islamic banking sector.

**Author Contributions:** Conceptualization, H.L. and W.H.; methodology, H.L.; software, H.L.; validation, H.L. and W.H.; writing—original draft preparation, H.L.; writing—review and editing, W.H.; visualization, W.H.; supervision, W.H.; funding acquisition, H.L. All authors have read and agreed to the published version of the manuscript.

**Funding:** This research was funded by Zhejiang Provincial Natural Science Foundation of China under Grant No. LGF22G030011.

**Institutional Review Board Statement:** Not applicable.

**Informed Consent Statement:** Not applicable.

**Data Availability Statement:** The data of this study are from the annual reports of listed banks.

**Conflicts of Interest:** The authors declare no conflict of interest.

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
