# Peer review of "Sustainable Financing and Financial Risk Management of Financial Institutions—Case Study on Chinese Banks"

_sustainability, doi:10.3390/su14159786_

Round 1
Reviewer 1 Report
I have the following comments:
1) The aim of the article is too general. The authors wrote "This paper aims to study the relationship between sustainable financing and the financial risk management of financial institutions", but the study concentrates only on banks in China, thus the aim should be rather as follows (for example) "This paper aims to study the relationship between sustainable financing and the financial risk management of financial institutions on the example of Chinese Banks" or "...and the financial risk management of Chinese Banks".
2) There are no research hypotheses.
3) Description of the data from Table 1 is on page 7, whereas table 1 is on page 4. The description should be a close (for example below) table (in this case 1) - it concerns also other tables and graphs.
4) Page 9 - why do the authors relying solely on the literature review concluded that "There are significant positive effects of banks' sustainable financing on their financial risk management, while there are significant positive effects of banks' financial risk management on their sustainable financing."? A significant positive effect should be confirmed statistically (using some statistical tests), not by literature review. Besides the author did not put forward research hypotheses, so why do they formulate such conclusions?
Reviewer 2 Report
The manuscript must be inspected by an editorial service. A lot of sentences are difficult to understand. ‘The Study on’ – remove from the manuscript title that should be also reorganized as finance-related terms are used 3 times. There are extra spaces between words throughout the manuscript. There are also missing spaces: Campiglio,2016, Heinemann,2006, etc. ‘Minor literature’ – if you mean non-peer reviewed, it should be ignored. If ‘a small part’ is intended, clarify this. ‘sustainable financing of banks and its financial risk management effects (FSME)’ – I don’t understand what the acronym stands for. ‘The sustainable financing amount of Big Four had been raised from RMB 3 trillion to RMB 5.6 trillion from 2016 to 2020, it increased by 25.7% year-on-year in 2020, and its proportion in total loans is 8.7% in 2020’ – updates are needed. ‘In 2021, the Big Four underwrite "carbon neutral bonds" to provide’ – refer to the past (underwrote) or use current position and data. ‘GLP’ – define it. You should compare your results with others in terms of concrete data for better research integrative value. ‘Liquidity ratio (CR)’ - CR? Why not LR? Figures need updates and specific analyses. The main contributions of the paper should be presented as part of the empirical discussions or critical assessment on the core research outcomes. Figure 2 is poorly designed with segmented words and text over picture. A Discussion section is missing. The conclusions drawn are not well justified in the data collected. Please provide more details regarding the study limitations and strengths and what this means for the study findings. The reference list does not follow the journal’s style, and some sources are incomplete. Why citing the non-peer reviewed version:
[5]. Bauer, R., Ruof, T., and Smeets, P. (2019). Get Real! Individuals Prefer More Sustainable Investments. Available online: 579 HTTPS ://papers.ssrn.com/sol3/paper s.cfm?Abstr act_id=32874 30 (accessed on 1/4/2021).
Instead of the final one:
https://academic.oup.com/rfs/article/34/8/3976/6237929
The relationship between green financial behavior and sustainable finance as regards financial risk management has not been covered, and thus such recent sources should be cited:
Ionescu, L. (2021). “Transitioning to a Low-Carbon Economy: Green Financial Behavior, Climate Change Mitigation, and Environmental Energy Sustainability,” Geopolitics, History, and International Relations 13(1): 86–96. doi: 10.22381/GHIR13120218.
Morales, L., Gray, G., and Rajmil, D. (2022). “Emerging Risks in the FinTech Industry – Insights from Data Science and Financial Econometrics Analysis,” Economics, Management, and Financial Markets 17(2): 9–36. doi: 10.22381/emfm17220221.
Kliestik, T., Valaskova, K., Lăzăroiu, G., Kovacova, M., and Vrbka, J. (2020). “Remaining Financially Healthy and Competitive: The Role of Financial Predictors,” Journal of Competitiveness 12(1): 74–92. doi: 10.7441/joc.2020.01.05.
Ionescu, L. (2021). “Corporate Environmental Performance, Climate Change Mitigation, and Green Innovation Behavior in Sustainable Finance,” Economics, Management, and Financial Markets 16(3): 94–106. doi: 10.22381/emfm16320216.
Reviewer 3 Report
The paper is well structured, the methodology rigorous, but I noticed that among the bibliographic references there are not many recent contributions. I therefore recommend supplementing the bibliography with studies like these from 2022:
Vagin, S. G., Kostyukova, E. I., Spiridonova, N. E., & Vorozheykina, T. M. (2022). Financial Risk Management Based on Corporate Social Responsibility in the Interests of Sustainable Development. Risks, 10(2), 35.
Landi, G. C., Iandolo, F., Renzi, A., & Rey, A. (2022). Embedding sustainability in risk management: The impact of environmental, social, and governance ratings on corporate financial risk. Corporate Social Responsibility and Environmental Management.
Kharlanov, A. S., Bazhdanova, Y. V., Kemkhashvili, T. A., & Sapozhnikova, N. G. (2022). The Case Experience of Integrating the SDGs into Corporate Strategies for Financial Risk Management Based on Social Responsibility (with the Example of Russian TNCs). Risks, 10(1), 12.
I would also reword the title to make it a little more catchy.
Round 2
Reviewer 2 Report
Some journal titles are not properly cited. E.g., here correct is: Econ. Manag. Financ. Mark.
14. Ionescu, L. Corporate Environmental Performance, Climate Change Mitigation, and Green Innovation Behavior in Sustainable
Finance. Econ. Manag. Financ. 2021, 16, 94–106.
22. Morales, L.; Gray, G.; Rajmil, D. Emerging Risks in the FinTech Industry – Insights from Data Science and Financial Econometrics
Analysis. Econ. Manage. Financ. 2022, 17, 9–36.
Please check all the cited sources again.